# Depth Psychological Elements in Seon Master Daehaeng's Dharma Talks, with Special Reference to *Hanmaum Yeojeon*

Chae Young Kim 

Department of Religious Studies, Sogang University, Seoul 04107, Korea; chaekim@sogang.ac.kr

**Abstract:** This essay attempts to approach the dharma talks of Korean Seon Master Daehaeng (1927–2012) from a modern scientific perspective. In particular, it tries to articulate depth psychological elements which belong to or which are relevant in some way to her dharma talks. In so doing, it will attend to the content of her *magnum opus*, *Hanmaum Yeojeon* (The Principle of One Mind), which was compiled from her extensive dharma talks. This essay articulates that she could be regarded in contemporary Korean Buddhism as a pioneer, the author of the first works which can be only understood properly if one's point of departure is the kind of meaning revealing depth psychological elements.

**Keywords:** depth psychological elements; religious experience; Seon Master Daehaeng; *Hanmaum Yeojeon*; transitional object; ultimate reference point; Juingong; Appa; subliminal unconscious

## 1. Introduction

Seon Master Daehaeng (대행; 1927–2012) was a Korean Buddhist nun. She is regarded as one of the most influential seon masters in contemporary Korean Buddhism. In 1972, she founded Hanmaum Seon Center (한마음 선원) in Anyang City, in South Korea, which eventually became one of the largest temples belonging to the Jogye Order (itself the largest order in Korean Buddhism). It has 15 branches in Korea but 10 branches in other parts of the world. After founding her center, Daehaeng pushed for a radical reformation of Korean Buddhism in order to modernize it and to encourage its global expansion. As one aspect of her work, she attended to how a contemporary scientific perspective can be properly applied to discuss any given question.

Daehaeng would leave a large archive of recorded materials from her dharma talks. Currently, most of the published works are based on her dharma talks delivered at the Hanmaum Seon Center to different meetings and groups. Among her published works, her *magnum opus* is *Hanmaum Yojeon* (The Principles of Hanmaum (One Mind)), which comprehensively recounts over more than 800 pages both the story of her life and the course of her religious experiences. Although more materials should be published in order to further our grasp of her thought in its entirety, for the sake of furthering a depth understanding of her underlying concepts, we cannot do better than to attend to that she says in her "Principles of One Mind". Currently, it exists as our best source if we are to expand our understanding of Daehaeng's life and the significance of her labors.

This essay does not refer to Buddhist interpretations of Daehaeng's dharma talks. Instead, it tries to situate her dharma talks in relation to sources which point to elements of depth psychology. In accompanying *Hanmaum Yojeon* as a key source of insight and information, it seeks critically to examine the events that were constitutive of Daehaeng's awakening, where this refers to the emergence of her religious experience and its ultimate point of reference in terms of depth psychological elements. In doing so, first of all, we will attend to the questions which Daehaeng was asking within the context of her life formation, moving from childhood years and beyond. Then, we will propose a psychological interpretation. We will focus on the meaning of her context and environment. Thirdly,

we will speak about the ultimate point of reference that was so basic for her as a point of transition in how she was able to understand herself and her world. Fourthly, we will look at depth psychological elements as these can be applied to her experience of Juingong. In conclusion, we will speak about how Daehaeng should be interpreted within contemporary Korean Buddhism in relation to depth psychological elements.

## 2. Questions and the Construction of a "Playground" in Childhood

Daehaeng's questions, beginning in childhood, were very crucial for her in awakening a deeper understanding of things that lead to her. Without these early questions about the meaning of life and death and the contradictions which exist when we think about life and death, it would be impossible for us not only to understand the context of her religious experience but also how she was able to expand and build her playground in her sense of life in the wake of her religious experience. Prior to or after her childhood religious experience, she had begun to ask questions with respect to how she could begin to relate to other persons and objects in a way through engagements with them, which serve, for us, as crucial points of entry if we are to understand the import and purpose of her life and the formation of character which resulted as an effect and consequence.

### 2.1. Questions

According to Daehaeng's *magnum opus, Hanmaum Yeojeon*, her life, from early childhood on, was filled with many questions (Daehaeng 1993, pp. 22–25) that are crucial to identify and to understand if we are to move toward a depth psychological understanding of her later works. In most academic research that has existed about her works, unfortunately, the questions that she had had, prior to her experience of Appa, seem not to be attentively grasped and understood. Most researchers and even many Hanmaum Seonwon devotees have tended to avoid thinking about the importance of her early questions. Unfortunately, her questions were usually seen to exist as instances of personal life stories that, later, were put into the shade by her later experience of enlightenment.[1]

However, this misconception needs to be critically reconsidered since the questions that led to her Appa experience are no less important than the questions which later emerged after her Appa experience. The two sets of questions should not be separated from each other since a mutual complementarity can be found between them that facilitates a better or a deeper understanding of Daehaeng in the context of her life and works. Most important is this the case if we should attend to her early childhood questions when, with her family, she was forced to live in a hut in a mountainous area. Namely, it should be attentively considered that questions emerging after her Appa experience cannot be properly understood apart from these early questions. Conversely, without later questions, her early childhood questions can be easily discarded or so easily interpreted as if they exist as pathological realities that are to be explained on the basis of the dysfunctionality which belonged to her family environment.

Hence, revisiting Daehaeng's childhood life is extremely crucial for us, although little remaining material survives outside her published recollections. Although she was born into a stable family environment, at age 7, she had to join her family in fleeing their urban residence to escape from the efforts of the Japanese Imperial police to arrest her father. (Daehaeng 1993, p. 22). Her family soon lost everything, their residence included, and they were driven to settle in a mean hut near a mountainous area. In this situation, her father changed in his character and he soon began to express his anger and to vent his frustration on Daehaeng, his hapless eldest daughter. In addition, she soon found that she had to do many daily chores. Even at nighttime, she found that she could not rest since her father would ask her to go down to a residential area to buy tobacco. For her, these nighttime

---

[1]　Since the present writer developed an interest in Daehaeng's life and works, he had several opportunities to discuss them with various people in and out of Hanmaum Seonwon. Even among her disciples, he could not sustain a serious discussion about the questions she asked in her early life. After presenting a paper on her life and works in 2019, he was challenged by a participant for focusing so much on her early life prior to her Appa experience.

trips, back and forth in the darkness, were very fearful experiences, all for the sake of running errands for her father.

As a consequence of these demands that her father was making on her, Daehaeng's parents often quarreled. Young Daehaeng came to believe that the main reason for these quarrels was her existence in the family, and so she decided to spend most of the night in the nearby mountain areas. This turned out to be the first occasion on which she left the family "home (hut)". She began to journey into a world of wild things in nature, noticing the various things within it: the grass worms, the trees, and the stones, and also the sound of the wind and the sound of animals and other things. As a young child, she was initially filled with fear. However, as time went by, the more nights she spent with wild creatures in nature, the more she began to relate and engage herself with these unfamiliar, unknown things. She ceased to be afraid of them, and even interacted with the gravesites she encountered on her journeys (Daehaeng 1993, p. 23).

*2.2. Character Formation*

Daehaeng's earlier life prior to her experiences of Appa/Juingong is crucial in terms of how it relates to her character formation. We can find two important qualities of her character which informed her life and works in prominent ways since her childhood.

The first was her massive capacity for empathy. Daehaeng, from an early age, manifested such a capacity for empathy that it enabled her to transform unfamiliar and unknown beings around her into familiar and known beings which surrounded her. In her case, it was not limited to only human beings but extended to all beings, whether in the visible or the non-visible world. Through varying emphatic engagements with them, she could establish new relationships with them and name them even as living "friends" (Daehaeng 1993, p. 23). She ceased to exist in an exclusive way as a separated being, with a distinct status which set her apart from the being of all the many things which surrounded her. Through her empathy, she could revive the dead. Shy, frightened living beings could begin to enter into a relation with her, and through a kind of familiarity which arose, familiar beings became as "friends" to her. In doing this, she frequently used to stare at many beings for a long time until she could know them in terms of the details in their surroundings with her (Daehaeng 1993, p. 24). Hence, her empathy could be identified as a patient type of capacity that could keenly penetrate and move into depth dimensions of being until, in some way, an invisible dimension that existed within things could then be grasped and understood.

The second was the constructive image or a sense of the "democratic playground" that existed within her since childhood and her life in the mountainous area near her family's humble home. She did not live or exist as an aloof being within a cold "hierarchical" construction separated from other things. Instead, she perceived them all "equally" ( Daehaeng 1993, p. 27); "equally", they exist as other beings who, in their own way, contribute to the construction of her playground in nature, irrespective of their individual situation or status. There were so many beings that she engaged herself with in her early years of mountain life and through a form of construction which, in general, divided into two main groups. (Daehaeng 1993, pp. 27–28). One shows itself through a kind of living presence which belongs to such things as wild animals, snakes, worms, ants, grasses, trees, stars, the moon, the sun, rocks, stones, and so on. The other shows itself through a connection to the lifeless or deathlike presence of dead or dying wild animals, worms, snakes, trees, grasses, and tombs. Like the first group, she equally treated beings of the second group. She perceived them as equal beings in the context of building and constructing her life's playground.

In this second case, she began to realize at the tender age of nine not only the vital reality of suffering in their being and existence but also, at the same time, she expressed her compassion to them and for them with heavy tears as if they belonged to one of her family members (Daehaeng 1993, p. 27). Although she experienced great trials in her childhood years, she chose not to live passively within this state, preferring to actively

engage with the experiences that were given to her in a way which transcended the circumstances of her personal life. Hence, beyond her personal experience, she began to ponder and think about the many trials and sufferings which exist in relation to the incidence of poverty, sickness, death and dying, and so on. She frequently pictured or imagined something which existed as an "invisible hat" (Daehaeng 1993, p. 25) and how it relates to the healing of our human suffering. At the same time, she often wondered and thought about many human existential problems and about what can be found or known about a fundamental reason which accounts for the existence of these problems. She ceaselessly searched for any possible resolutions. She often used to display and expose her strong will at a young age, so much so that, if she could not get an answer, she felt it would be better to die (Daehaeng 1993, p. 28). This type of dynamic, in her life, continued to function as a principal force that propelled and carried her throughout the rest of her life, pushing her in her journey in the quest or the search that, for her, was always central.

## 3. The "Psychological" Playground in the Years of Childhood and Beyond

For the purpose of academic research with respect to the matter of Daehaeng's works, psychological perspectives such as Jung's analytical psychology, humanistic psychology, transpersonal psychology, or object relation theory will be one intriguing approach, among other candidates[2]. Especially in this section, object relation theory could be appropriated for the psychological interpretation of the playground.

The image that we have of Daehaeng's playground will, in general, point to explicit characteristics that will refer to the kind of clinical or psychotherapeutic space that exists between an analyst and an analysand, a psychotherapist and a client, a counsellor and a counselee, or a doctor and a patient. Most especially, her notion of playground resembles or, more strongly, is compatible with the notion of playing place as a species of "transitional object" (Winnicott 1951)[3], which Donald Winnicott (1896–1971) has described in his major book, *Play and Reality*.[4]

With respect to this point, to clarify Daehaeng's notion of playground in relation to the space of play which belongs to a transitional object, we should ask one important question about the place, as the *locus* of her playground, in mountainous areas near where Daegaeng had lived. Is it located in the outside objective world, apart from her mind? Or is it located exclusively within her subjective world? Her emphatic engagement with the many objects that she encountered in the mountainous space and area of her childhood seems to indicate that its place is not restricted to either an objective or a subjective world. It is to be found rather within a kind of inter space which exists between these two worlds. It is that which emerges from how, sympathetically, she had encountered and engaged with other beings within the world of nature which existed about her and this notion of space is identical to the place of play with respect to the existence of transitional objects.

According to Winnicott, three things are fundamental in the formation of the place of play for a transitional object. They are: (1) the human subject, (2) an object which is the focus of one's engagement, and (3) the surrounding world or environment (Winnicott 1971, pp. 1–3). For example, in the case of a baby: (1) a baby, (2) his/her mother's breast or feeding bottle, and (3) his/her surrounding environment. A blanket or swaddling clothes exist as key components in forming a place of play (in this context). A baby's repeated relations with these eventually create a place of play where all elements exist as participants. A baby

---

[2] This tendency, in other regions beyond the peninsula of Korea, would be exclusively found in relation to Zen Buddhism (Kato 2005, 2016; Kim 2012).

[3] Donald Winnicott coined originally the term *transitional object* in his talk under the title "Transitional Objects and Transitional Phenomena: A Study of the First Not-Me Possession" in the British Psycho-Analytical Society, 30 May 1951. Twenty years later, he articulated it in his well-known work, *Playing and Reality* (Winnicott 1971), within the frame of transitional phenomena.

[4] According to Winnicott, we become really and truly human through being involved with a multitude of transitional objects, as, for instance, in our very early life: a mother's breast, a baby's feeding bottle, and a child's teddy bear (Winnicott 1971, pp. 8–11). This feature of transitional objects repeatedly appears in our later years of life as we move from youth into adulthood and then into our old age. Although their contents in later life differ from what had been given to us in our earliest years, the pattern of transitional objects in our later life expansively repeats how, in one's early years, one had been related to a multitude of transitional objects.

would understand, in its own way, the other two components initially in a partial way but, after some period of time, the baby would understand them and their reality entirely within the relevant space of the child's play. Hence, a place of play is not simply formulated, apart from the being of transitional objects, whether they are visible or invisible. It is formed within the context of an inter space which exists between a human subject and any given transitional object (Winnicott 1971, p. 108).

In this sense, Daehaeng did not create her playground in an entirely subjective way, apart from the interjection of other beings and the influence of any surrounding mountainous environment. What exists in fact is a form of ongoing constructive process that emerges as all three components relate to each other. For Daehaeng, her playground emerged because of a role which she herself played: she emphatically engaged and involved herself with the other two components in a way which situates her place of play within a larger context which refers to the harshness of her earlier life—the cruelty of her father, the dire poverty of her family, and the weight of other existential matters (Daehaeng 1993, pp. 28–29). And so the place of play becomes a place of healing within which all beings undergo a kind of revival which makes them which turns them into new beings. She raises them up the constitutive parts or elements in a positive way; they become full players in the context of her playground. No discrimination is known. No exclusion is practiced. All is equality and inclusion.

Like Winnicott's notion of playground, Daehaeng's notion of playground is not restricted to her years of childhood since it reaches into all the remaining years of her life in a dynamic expansion grounded on the archetypal playground that she had creatively constructed in her early years of childhood. As we have seen, this all began in an adjacent mountainous area near her family's hut/home, and only later was this process perpetuated in other areas and places. To expand the size of her playground, she began to live in other mountainous areas (Daehaeng 1993, pp. 74–76). Almost all the mountains in the peninsula of Korea became, eventually, another playground for her, where she would play with so many visible and invisible beings. Later, she looked back and expressed herself through one explicit statement which spoke about all beings and how they relate to the many mountains which she had visited—all of these which had been transformed into playgrounds must be "Tripitaka Koreana," the comprehensive collection of woodblock Buddhist scriptures and texts (Daehaeng 1993, p. 66). For her, everything that she had encountered by the mountains that she had known and beyond exists as an exemplification of Buddhist teaching and scripture (Daehaeng 1993, p. 75). In the same way, for St. Augustine, we recall how he spoke about the presence of two scriptures in Christianity: the Bible and the surrounding world of nature (Chardwick 1992).[5]

Daehaeng's emphasis on the place and role of invisible Buddhist scriptures in nature does not mean, however, that she totally rejected the study of scripture for the sake of enjoying material blessings. Instead, she harshly criticized a blindly literalistic study of Buddhist scripture when it was combined with an expectation of material benefits if there exists no critical or experiential reflection about one's self and who one is as a human being (Daehaeng 1993, p. 166). Hence, after her experience of enlightenment, she gave serious thought to the need of establishing a context that would help other persons in a way that would further their education in a reasonable and a responsible way.

In the wake of her experiences and realizations, she began to expand her notion and sense of playground in four distinct ways. First, she expanded her playground to include an urban, residential area. In doing so, she resided in a small empty crypt which was named as the "Gyeunseong hermitage." It was located under the Sangwon temple, adjoining the mountain of Chiak (Daehaeng 1993, p. 95) and, in this place, she received many requests from other persons who were seeking her help and healing. From this time on, she became more involved with other fellow human beings and their life

---

[5] St. Augustine dealt with the world of nature as the other scripture in addition to the Bible in his last three books (XI–XIII) of *Confessions* on the examination of Genesis. He also expansively made commentary on Genesis in *Literal Commentary of Genesis.*

struggles. Second, she established the Hanmaum Seonwon in Anyang in 1971 as an urban playground where many people could fully participate as new beings at age 40 and beyond (Daehaeng 1993, p. 103). In addition, she expanded her playground to include other urban playgrounds beyond Korea, moving to other regions in a way which pointed to a form of global mission and outreach which she sought to implement and bring into being (Daehaeng 1993, pp. 138–45). Third, she expanded her playground to include a cosmic or an invisible world. No unknown thing is to be excluded but, instead, be actively embraced (Daehaeng 1993, p. 145). Fourth, she attempted to establish her playground in a way that was rooted in a human subject's concrete body (Daehaeng 1993, p. 255). She identified the world of the body as a living playground in which, in this world, many different things participate as equal players whether be they organs, cells, or other living things.

### 4. The Playground and Its Ultimate Reference Point

In this section, we would articulate three aspects of the playground which Daehaeng constructed from early childhood onward: constituent elements, the "more" dimension, and religious life transition.

#### 4.1. Constituent Elements

Distinctively Daehaeng emphasized two constituents of playground over the later course of her life. The first concerns the necessity of having a firm faith. For her, as Corbett notes (Corbett 2011, pp. 72–74; Corbett 2015, pp. 103–18), religious life does not exist as some sort of magic, apart from our need to be personally active and involved. In this respect, she harshly criticized all forms of magical thinking that can occur in religious life. She firmly believed that, through a form of ongoing practice, religious life should exist as a dynamic form of journey. However, the journey is inevitably multifaceted, sometimes quite peaceful or joyful, and at other times harsh and uncomfortable. Especially within the context of some emotional or psychological depressions, one can be tempted to abandon one's religious faith. Hence, to avoid this, Daehaeng always advised her followers to not avoid bad situations but, instead, to actively embrace our trials with a firm faith (Daehaeng 2014, pp. 116–18).

This emphasis on the importance of a dialectic dimension accordingly provides a new perspective which is able to embrace opposite sides if now, we are to engage in a critical reflection about the meaning and practice of religious life and so come to its ultimate foundations as this can be found in the life of Daehaeng in the wake of her first experience of Appa in terms of its meaning and being. In fact, if we should speak about dialectical experiences, a dialectic experience was given to her in terms of her "religious conversion" (Lonergan 1971, pp. 251–53; Kim 2010), which rejected the conventionality or the reification of Korean kinds of Buddhism. As a result, thus, she was able to establish a new religious foundation, prior to communicating her works in published writings to her many followers. This is the main reason that her works continue to be "reasonably" appealing not only to her followers but also to diverse peoples beyond the boundary of conventional Buddhism.

The second concerns the emphasis that she gave to the importance of the experiential dimension as one's ultimate, fundamental point of reference. As we noted earlier, Daehaeng's religious concern in relation to Appa differs from the current type of approach that we tend to find within the context of academic research and the type of religious environment which is often to be found in temple worship and observance. As a point of contrast, she refrained from trying to comprehend her own experience in a way that would be rigidly grounded in doctrinal or speculative issues as we find this within the texts of scripture and their assorted commentaries. Instead, she sought to comprehend things in an experiential way and, as a result, her work became quite unique within the contemporary stream of Korean Buddhism. The emphasis that she gave to the importance of experience drove her to reorient or to creatively revise the Buddhism of her day with respect to its study and practice. In this sense, more so than anyone else, she emphasized

how the experiential dimension within the life and practice of contemporary Buddhism is, in fact, more important than the good of meditation and healing as a phenomenon that was also emerging within the life of contemporary Korean Buddhism.[6]

William James distinguished between two types of religious experience, sudden conversion and gradual preparation (James [1901] 1929, pp. 163–85; Kaag 2020). If Daehaeng's early years were to be ignored and put to the side, then her experience of Appa would be seen as if it were a representative example of a sudden type of religious experience. But, if her early years are viewed in terms of its being a time of preparation, then her experience cannot be viewed as if it were a sudden type of religious experience. Instead, because we find elements of preparation in her early years which led to her conversion, her experience can be identified in a way which sees it as a representative example of a gradual type of religious experience and conversion. However, as a consequence of this tendency, the story of her early life has been unfortunately discountenanced. It has not been seriously integrated into how we are to understand her depth experience of enlightenment, although, simply put, this is a major misunderstanding that is still adhered to by many of her followers and devotees and by most scholars.

### 4.2. The "More" Dimension

On the basis of these two constituents (faith and experience), Daehaeng developed a sense of her ultimate point of reference, which was embodied in the sense that she had of her playground. She did this through diverse images and acts. Although, within her, she was endowed with an empathic capacity that extended to all beings as her "friends", initially, she did not have full knowledge about what exactly was her fundamental point of reference. She initially only felt and imagined that which existed as the "more" dimension of her playground before she could reach out fully to an enlightening form of concrete knowledge. In her early childhood years, she was not without a strong feeling of potentiality with respect to this "more" dimension. The more she engaged in a religious form of wondering in adjacent mountainous regions in Korea, the closer she moved toward an encounter or an experience which was able to transform the potentiality of her "more" dimension into an actuality that could come to exist in terms of cognition and knowledge. The emergence of this knowing was, however, not accomplished suddenly but through a long and gradual journey that began for her from the time of her childhood.

In her life, on several occasions that were both pivotal and foundational, her potential feeling and imagination underwent changes in shifts which moved into instances of actual knowledge, although, at every step along the way, her experiences and receptions of understanding and knowledge were never exhaustive. An openness of a "more" (James [1901] 1929, p. 500; Kaag 2020) dimension existed. Nevertheless, however, the ultimate point of reference which grounded her notion and sense of playground can be identified, in her works, in terms of three desires, occasions, orientations, or orders of intentionality that encouraged her to move toward a knowledge of this "more" dimension within an awareness that this "more" dimension is something "plus" (James [1901] 1929, p. 489) that can never be entirely and fully known by her or by any of us.

This quest would require successive orders of intentionality. The first order of intentionality that was oriented toward acquiring some degree of understanding and knowledge originated from a direct experience of Appa that erupted from within the inner depths of her mind when she was only nine years old (Daehaeng 1993, p. 29). After this first experience, she began to know that Appa exists as the principal source of consolation in her life (Daehaeng 1993, p. 29). In the years to come, she would try not only to re-experience the consolation of Appa but also to rearrange all other beings including herself in lieu of this early experience (Daehaeng 1993, pp. 29–30), although, visibly or materially, in the

---

6　This point is so fundamental that it allows us to be psychologically connected to the study of religious experience since the birth of the psychology of religion movement which dates from the early 20th century (Hood et al. 2009, pp. 20–22; Paloutzian 2017, pp. 40–50; Kim 2010, pp. 982–83; Kim 2016, pp. 1246–47).

circumstances of her later life, nothing differed from the circumstances of her prior life. What changed was her realization that she could no longer exist as her own center as, now, Appa became the new center of her life and playground. Prior to her first experience of Appa, the *locus* of Appa existed only as a potentiality but, after her first experience of it, it became a new center in a displacement which marginalized her past place and condition. This change, as the most important moment of awakening for her in her life, encouraged her to try and move toward a more intimate knowledge of this "more dimension" as this refers to the being and reality of Appa (Daehaeng 1993, p. 30).

The second order of intentionality that was oriented toward an understanding and knowledge of Appa can be found in a later, deeper form of religious quest that ensued after her initial experience of Appa. In this period, Daehaeng became the subject of a number of dynamic experiences that came to her in the wake of her unlimited desires to know Appa. Whether these experiences were given to her in mountainous or urban areas, despite variations in location, she was carried and sustained by an underlying desire to know Appa much more fully than what had been given to her initially. In so seeking and yearning, surprisingly, she stayed for some time at a Catholic Church and thought about possibly becoming a nun (Daehaeng 1993, p. 34). She also stayed at Buddhist temples in seeking a greater knowledge of Appa and also met a number of monks (Daehaeng 1993, p. 34). However, she could not find the consolation that she desired that, in some way, resembled the power of her initial experiences (Daehaeng 1993, pp. 35–36). At this time, she met Hanam Sunim in person and, from him, received teachings and advice about the possibilities and probabilities of moving into an in-depth study of mind, although, at the time, she did not clearly or fully understand the meaning of it (Daehaeng 1993, p. 38). At that time too, at a personal level, she came to regard Hanam Sunim as a personification of Appa because of the comfortable feelings which she had with him, as if he were a grandfather to her (Daehaeng 1993, p. 40).

*4.3. Religious Life Transition*

In Daehaeng's quest to know Appa, three stages or three dynamic instances served as points of transition in her life. The first was an experience of religious thirstiness when she was 18 years old. She could not stay comfortably at home because of her mother's obstinate requests that she should get married (Daehaeng 1993, p. 41). However, apart from her experience of religious thirstiness and her desire to experientially know Appa, she indicated that she was quite uninterested in getting married (Daehaeng 1993, p. 40). Instead, she would frequently go to nearby hillside areas. When alone at home, she would experiment: standing in front of a mirror, she would try to come to a knowledge of Appa (Daehaeng 1993, p. 42), frequently waiting for several hours for an answer, writing to Appa in words which said that "if, Appa, you exist, you should do it." She would similarly stay overnight at a temple close to a national cemetery (Daehaeng 1993, p. 43). However, the more she engaged in these experiments, the more her uncomprehending mother would press her daughter to get married. After a time, Daehaeng decided to leave her family home.

The second instance or stage was the time of her success in secular life when living in Busan and then returning to an earlier concern: the work of *bosalheang* (보살행, compassionate service) to help poor laborers. After leaving home, she moved to Busan, where she opened a small repair shop and restaurant. While she was economically successful in her business, her desire to know Appa was not fulfilled. Eventually, she tried to actualize her wishes in terms of trying again to help needy persons who were living in mountainous areas, as she had done prior to her experience of Appa. In the end, however, she could not maintain her life in Busan, given an unceasing desire to grow in her knowledge of Appa. She left Busan and returned to her forest life again in order to pursue her religious quest in adjacent mountainous areas. During this time, when back again in forested regions, she cried out to Appa, with a strong faith, that she wanted to encounter Appa and to let Appa

do everything in her life (Daehaeng 1993, p. 46), although, despite her desires and devotion, she could not be liberated from her religious thirst, from her desire to know Appa.

The third instance or stage began with her life as a novice in a Buddhist temple. After living in a forest, she eventually entered the novitiate at a Buddhist temple. She cut her hair and she continued to yearn and search for a knowledge of Appa (Daehaeng 1993, pp. 47–48). However, because she asked many critical questions about the meaning of Buddhist religious life in her new temple home, she found that she was soon misunderstood and was even thought to be insane. As a result, she was eventually evicted from the temple (Daehaeng 1993, pp. 49–51). After this rejection, she again returned to forest life. Eventually, she went to the Sangwon Buddhist temple in the hope of meeting Hanam Sunim. She was fortunately able to meet him there as well as his disciple Tanheu Sunim. Under Hanam Sunim's direction, she eventually became a monk on 27 March 1950 (Daehaeng 1993, p. 53).

More intensely, in this third instance or stage, Daehaeng came to know Appa as her ultimate point of reference after she officially became a Buddhist monk. Although she was initiated to be a monk, officially, she did not belong to any temple. She continued to live in a forested area for the sake of her religious quest in order to come to a fuller knowledge of Appa (Daehaeng 1993, pp. 55–56). However, she often had to return in the daytime to care for her family and run a small shop. At night, she would return to her forest home, where she continued to practice her religious quest. At this time, her desire to come to an experiential knowledge of Appa became so powerful that she could not continue to ask questions, such as "what is it to live?, who am I?, or why can't I meet Appa?"(Daehaeng 1993, p. 57). Sometimes, she would not be able to eat for several days and nights and she would repeatedly hear within the same answer that was given to her questions: "if you were to die, then you would see me" (Daehaeng 1993, p. 57).

In the midst of these struggles and her unquenchable desire for an experience of Appa, she would literally interpret the answer that was given and, at times, she tried to die, attempting suicide several times in order to move toward an experiential knowledge of Appa (Daehaeng 1993, pp. 58–59). For the sake of her religious quest, she began to stay in steep mountainous areas that left her face and body scratched by wounds. On the rare occasion when she came down into a residential area, she was seen by many people, including children, who mocked her, as if she were a mad woman (Daehaeng 1993, pp. 60–61). She also continued to stay within the area of an adjacent national cemetery, where she would engage in a dialogue with various dead spirits (Daehaeng 1993, pp. 62–63). At this time, she began to comprehend more about how compassionate help exists in a mutual way among many living beings within nature, and she came to realize that everything exists in its own way as a living form of scripture.

At this time, Daehaeng began to experience a degree of rest with respect to her burning desire to have more experiential knowledge of Appa. She would have experiences of Appa by merely staring at two large tombs where, respectively, someone's father and his son were buried (Daehaeng 1993, p. 78). She speculated that if the father were to come to his son's tomb, he would become his son and, conversely, if his son were to come to his father's tomb, he would become his father (Daehaeng 1993, pp. 78–79). In her experience, they did not exist as separate beings but one being given the kind of crossover that was not existing between the father and the son. Within this experience, she could begin to know that Appa did not dualistically exist as some kind of separate thing occupying a specific place but as *Hanmaeum* (One Mind) that was instantly given through a form of unlimited interconnection which exists with respect to all things, whether they are visible or invisible.

In the context of these new experiences, Daehaeng could confirm that the Appa which she had initially experienced and which she was attempting to re-experience in her life could be momentarily "herself," as her being slipped the tether of any particular (Daehaeng 1993, p. 79). According to this confirmation which was given to her, she could even change the name of *Appa* to *Juingong*, where Juingong refers to that which can ceaselessly and momentarily occur in any place, at any time, without the need or necessity

of there being any fixed, centering place. Hence, in the absence of a need for any fixed place, Daehaeng could emphasize the meaning of the last syllable, "gong", in the new name where, now, within this context, Juingong distinctively points to what exists as a meaning for emptiness. In this sense, the only compatibility in terms of literal meaning which exists between Appa and Juingong is restricted to the significance of the syllable "Juin." The difference in denotation perhaps best explains why, at this time in her life, she was able to change the name of the ultimate point of reference for her playground.

## 5. The Presence of Depth Psychological Elements in Juingong Experience

In the wake of her enlightening experience of Juingong, we can find that Daehaeng's works tend to display a number of depth psychological insights (Kim 2019, p. 274) in relation to her ultimate point of reference, which grounds the perspective of her playground. Although she was not trained in psychology, she was quite well versed in the terminologies that belong to the field of so-called depth psychology. Sometimes, in her elaborations, we notice her direct usage and quotation of depth psychological terms. For example, she often used such depth psychological terminologies that refer to the subliminal, the subconscious, and also the unconscious (Daehaeng 1993, pp. 401, 544) when, in an expert way, she would talk to her devotees about the reality of Juingong. If we attend to the meaning of her usage, not much difference can be found if we compare her usage with how the same terms are used in the context of the kind of discourse which belongs to depth psychology.

However, we should not assume that her usage bears the same implications as those that exist within the different branches of study that belong to the field of depth psychology. She seemed to be aware of these differences with respect to how the meaning of terms differs as one goes from one school to another. Quickly is this point grasped as soon as we realize that diverse schools in depth psychology work with different nuances of meaning that are reflected in how certain words and concepts are used within differing contexts. While, currently, we cannot find any specific material or a specific source that she could have used to borrow her manner of use and nuances of meaning, if we look back and revisit the modern history of psychology, we should soon realize that, in an indirect way, her usage and her meaning are things which can be aligned to some theoretical and practical approaches specific to the field of depth psychology. She carefully delineated this point in many of her talks. In doing this, she emphasized three things: life attitude, therapeutic function, and image in relation to Juingong.

### 5.1. Life Attitude

Daehaeng emphasized three attitudes of life for the encounter of Juingong. First of all, she emphasized the importance of confronting or facing our life problems instead of avoiding or repressing them (Daehaeng 1993, pp. 483–85). This emphasis does not come as a speculation that is derived from a conceptual order of thinking that allegedly exists in her thought. Rather, it comes from her life experiences that began prior to any of her encounters with Appa or Juingong. Without her experience of having to confront her feelings of fear, stress, boredom, and anxiety and without her experience of having to deal with visible and invisible beings prior to or after her the initial experiences of Appa, she could not have confidentially encouraged her many devotees to confront or also face their problems in a similar way within the metaphorical "mountains" or the "forests" in their own way within their lives.

Whenever she was able to talk to others, Daehaeng would express the fact that we all need to confront our problems as they relate to ourselves (Daehaeng 1993, p. 485). The manner of her speaking was akin to that which belongs to a piercing Seon master. She did not restrain herself in voicing harsh criticisms against "superstitious" or "magical" efforts that would fail us in any serious effort to confront the real issues in each of our lives. In addition, she strongly emphasized that nobody can deal with these issues other than by "oneself." This did not mean that we can solve our problems only according to the partial kind of reality which belongs to us if we should refer to our rationality and our ego and

the sum of our socially acquired skills that belong to each of us within the borders of our conscious life. She criticized efforts of this kind as always an idolization of one's partial reality where, falsely, we assume that it is our entire, full reality (Daehaeng 1993, p. 486). Hence, the emphatic type of meaning which she ascribed to "oneself" as if it were a reality that embraces the unconscious in one's psyche or mind beyond the conscious realm—this is something which differs from a subjective notion of consciousness which is widely dispersed within our modern world and which inflates the scope and the competency of our individual ego or that which exists as our individual human reason.

Secondly, Daehaeng emphasized the good of having an entrusting or a letting-go attitude in life if we are to deal successfully with life's many problems. Simply put, we need to turn things over to the life and being of Juingong (Daehaeng 1993, p. 487). We cannot think that we can solve our problems apart from grounding ourselves in Juingong since, without this, we would be left with only an unintelligible species of ultimate void. To reiterate: for Daehaeng, we need to entrust our problems to Juingong. However, the place or the role of Juingong in this context is not such that our trusting only points to a number of ideal or a number of angelic features which exist since, at the same time too, a kind of void with a number of demonic features is revealed to us (Daehaeng 1993, p. 370). Together, these two features are seemingly and strangely intertwined. However, in the context of her many talks, Daehaeng seemed to speak more about the being of a void and the being of demonic features if we compare her emphasis with how she spoke about the being and the quality of other features. To resolve our problems in relation to Juingong, she believed that we should somehow return to the original place or to the original origin of our problems as they have emerged from the demonic features that continue to exist within Juingong (Daehaeng 1993, p. 487). We cannot solve our problems and difficulties by ourselves within the order of our conscious psyche or the order of our conscious minds. Hence, we best proceed if, through our efforts and desires, we seek to transcend the order of our conscious life and being and so try to "return" to the original place of our problems if we directly confront or face them as they strangely exist within ourselves. More than anything else, this focus on return and on confrontation exists as key if, successfully, we are to deal with our many, seemingly intractable problems.

As a third point, no one can adopt a "vicarious" role in order to solve another's problems. One should be responsible for one's own problems and so, by oneself, place one's faith in Juingong. In working with her devotees and in trying to help them, she explicitly told them that she could not solve their problems for them apart from the advice to trust in the being, the goodness, and the reality of Juingong (Daehaeng 1993, p. 244). To make any progress here, one must break from one's typical or conventional attitudes in life if one is to move toward Juingong in a way that is not to be compared to some kind of act of blind faith. As she admitted to her followers and devotees, she could not vicariously solve their individual problems. She could only act as a servant, trying to guide her followers toward the place of Juingong by encouraging a kind of trusting faith in that which they should have within them (Daehaeng 1993, p. 150). To make any personal progress at all, each of us needs to attend to the master existing within each of us (Daehaeng 1993, p. 545).

However, while her frank statements about "herself", as she spoke to her followers, reject all forms of blind reliance that they could have on her, in addition, they reveal to us a form of humility that penetrated her life and works. Although, admittedly, among her devotees, many saw her as an authentic religious leader and they followed her in this regard, from a psychological perspective thus, we cannot understand her life nor can we understand her work as if it points to some kind of inflation of ego emanating from her charisma, as has been with the case with founders and leaders of religions, especially among new religious movements.

### 5.2. Therapeutic Function

More specifically, Daehaeng's three attitudes of life are connected to the therapeutic function of psychology as a means that can help us move toward a greater growth in

the degree of our self-knowledge. According to the insights of depth psychology and the teachings of classical psychoanalysis, our mind or psyche manifests symptoms of illness just as the body does, although, in terms of mental illness, we need to transcend any conditions or states of repression within and below our conscious existence (Corbett 2015, pp. 155–57). Through a form of psychotherapeutic conversation that exists between a therapist and a client, a counsellor and a counselee, or an analyst and an analysand, these repressions can be identified and so, in some way, released or expressed, although, in this process, resistance from a client, a counselee, or an analysand will impede further development and healing if, in some way, further conversations or dialogue are prevented or impeded. Conversely, an oversight on the part of a therapist, a counselor, or an analyst will lead to problems that will arise from misdiagnosis. Eventually, a bifurcated situation will tend to encourage a growth of uneasy feelings within oneself that will point to negative self-determinations about who or what one is as a human being—a perennial problem within the practice of psychotherapeutic encounters within the context of any form of depth psychology (Corbett 2015, pp. 155–72).

In the context of Daehaeng's life and works, we find that a therapeutic role exists if we look at how she related to her many followers and devotees. Her way of talking and listening to the different kinds of objects which belonged to the order of visible and invisible beings and, later, how she spoke and listened to the voices of her followers and devotees and others—these things resembled how psychotherapeutic healing skills are exercised and displayed. She seemed to consider many different components in a way which conditioned how effectively she could speak to other persons and how she could also attentively listen to others, noticing in herself and others how communication exists in a way that is not purely linguistic, formal, or denotative. She did not fail to note when, from her followers and devotees, there were forms of resistance which existed in them as they tried to evade identifying troublesome problems and difficulties that they needed to know about if they were to begin to address them and then begin to find proper, adequate solutions.

Daehaeng had a gift that protected her from the kind of misunderstanding in having unwanted oversights. As her disciples and devotees would often witness and testify, experiences of self-knowledge and insight would come to them through their private conversations with her or/and through hearing her public talks and lectures. For them, it was akin to an experience of healing in a "melting down" or a disintegration of hardened symptoms. Daehaeng would often compare this kind of experience to an experience of catharsis at the moment of defecation (Daehaeng 1993, p. 237). Later, in order to concretize her psychotherapeutic concerns and interests, she developed a congruent "science of mind" (Daehaeng 1993, p. 419), establishing a medical hospital complex that was devoted to the total care of patient symptoms in a way which could transcend the kind of care that typically belongs to biomedical models of medical practice (Daehaeng 1993, pp. 281–83). The result was a Korean version of depth psychology and how it can be applied within the context of a functioning medical hospital.

To her devotees and followers, Daehaeng was able to articulate her understanding of resolution and insight experience as this was related to her experience of enlightenment and to its depth psychological applications. As we have seen, she had an enlightening experience when she perceived the two large tombs of father and son. When, phenomenologically, the two tombs seemed to be separate, in their relation with each other, they existed as if they were one tomb. In this sense or in a similar sense, while we are phenomenologically separated, in fact, we are united to each other and so, in this way, she came to realize that Juingong cannot be stably separated from us or the being of all other things but, instead, we best speak about forms of union and limitless interconnections that exist among visible and invisible beings within this world and beyond. Our relationship with Juingong exists as no exception. We are one profoundly with Juingong and so Daehaeng would always ask her followers and devotees to attend to an awareness of this oneness. For the sake of better forms of communication with others, she would appropriate psychological language

in order to tell others about experiences of enlightenment and insight which came to her and which come to her through the encountering kind of experience which belongs to us in both our present consciousness of things and our subliminal consciousness of things (Daehaeng 1993, p. 579).

### 5.3. Juingong Image

Daehaeng described Juingong with diverse images whenever she would have occasions to deliver her dharma talks. Especially, she employed four images in relation to Juingong in those occasions.

The first image for Juingong refers to a furnace (Daehaeng 1993, p. 323; Daehaeng 2007, p. 77) as a place in which dirty, rusty scrap metal is melted down and converted into pure, clean metal. Juinsong is where the hardened "rusty and dirty" things of our minds are melted down and transformed as our subliminal, subconscious, unconscious minds come back as a new self. This needs to keep occurring if we are to maintain some kind of "pure, clean" mind. We cannot engage in any kind of change or "melting down" apart from our faith and so apart from bringing our ego to the place and furnace of Juingong. After bringing our ego to the place of furnace that exists within the context of our faith, we can then simply wait and behold the process as it touches us and occurs within us.

A second image refers to a gardener (Daehaeng 1993, p. 234) who carefully cuts away useless and dead things in order to maintain a garden. Sometimes, overgrown leaves and branches need to be pruned if a harmonious network of living parts is to be connected in a way which makes for the beauty of a garden. Like a gardener, Juingong cuts away at our overgrown branches and leaves, removing also dead things which are not properly connected if there is to be an interior form of garden which exists as a "pure, clean" mind. Here, also, Dahaeng emphasizes the fact that we exist as limited, partial minds; we cannot exist as our own gardeners, pruning our useless branches, as this exists within the garden of our minds. However, we have Juingong and Juingong exists as a personal force or power; or, in other words, the subliminal, the subconscious, or the unconscious exists as a kind of personified being. It can be interpreted in this way.

A third image is a telephone or a mailbox (Daehaeng 1993, p. 254). These two images are based on the relation between a sender and a receiver. If a relation cannot be formed, they would be useless. If a receiver is not able or ready to receive a sender's message, then it cannot be delivered. For reception or for communication, a receiver must be ready in a way which resembles a form of trusting faith where one allows one's life issues to come to one in an unobstructed way—in a way that points to the good of waiting and being patient. Sometimes, a receiver cannot wait for an answer that is supposed to come from a sender and so is tempted to come up with mythical answers according to one's own inclinations, illusions, or ego. The impatience of a receiver can possibly thwart hearing a sound which refers to an answering ring that is being sent by a sender. A telephone or a mailbox resembles the kind of place that belongs to Juingong when or where Juingong attempts to send an answer or a message to a receiver.

A fourth image refers to a washing machine or a master key in a factory (Daehaeng 1993, p. 274). Their function resembles that which belongs to a furnace since they exist as images of cleaning and producing. For the cleaning and producing, a form of trusting faith is necessary since, without being connected to these things, we cannot engage in any form of cleaning or producing. One must have faith or one must entrust one's faith to the kind of place which is imaged by a washing machine or a master key, wherever these things are located. In this sense, for Daehaeng, the most urgent requirement which must be met by her followers and devotees is the development of a firm faith with respect to a connection with the place or the context of Juingong. Hence, whenever she spoke about the place of Juingong, she would always encourage her followers and devotees to keep a firm type of faith.

With respect then to these four images of Juingong which Daehaeng spoke about, one common theme joins them. We can only properly function in our lives (Daehaeng 1993,

p. 489) if we unite our life problems with Juingong by the agency of our personal faith. In any given plant, an invisible energy ceaselessly flows for purposes of "melting down," "the cleaning and the producing", and this type of movement resembles a flowing form of energy in the realm of the subliminal, the subconscious, or the unconscious which exists in depth psychology. This energy exists in the nature of all beings as a life-giving thing, the nature of a given thing pointing to the kind of energy which properly belongs to it. Hence, if we are to renew the life of our psyches, we must attend or allow for a form of energy that is joined to foundations in the unconscious not unlike the perceptions of psychologists from F. W. H. Myers (Myers 1903, 1904) and Richard M. Bucke (Bucke 2011) at the turn of the twentieth century through the contributions of subsequent psychologists attuned to the numinous (e.g., Jung 1970, pp. 6–7; Jung 1995, p. 173; Otto 1958, p. 6; cf. Coe 1900; McDermott 1986; Wulff 1997, pp. 21–28; Kemp 1992; Kim 2010, pp. 983–84).

## 6. Conclusions

Until now, we have tried to clarify depth psychological insights in the life and the works of Daehaeng. In this process, we have come to know about her concern to use currently understandable communicable language in a way which could enhance the effectiveness of her message for her many followers and devotees. Rather than exclusively using Buddhist language, she actively appropriates language that comes from a number of current academic fields—most especially, language which is explicitly compatible in terms of meaning that belongs to the studies of depth psychology.

While Daeaheng did not receive any academic training in depth psychology, we can confirm that she seemed to know about the existence of different branches in depth psychology and, from this, she could speak in a way that appealed to her many followers and devotees. Nevertheless, we could realize that she carefully considered two things in how she spoke. First, the language of psychological reductionism is to be avoided since it focuses only on the realm of the ego and the being of a person's unconscious. Her psychological language with respect to how she speaks about the place of Juingong transcends the personal realm of each individual since it is not located in any specific place. It exists everywhere, in no particular place.

Second, she knew that the core energy which flows from within the depth dimensions of our mind or psyche, whether it is in, out, above, and below—this depth dimension must be religious, spiritual, or sacred. Unfortunately, modern psychology, as a consequence of its empirical methodological orientation, does not actively promote this dimension, even if we should want to move toward a deeper understanding of the human mind or psyche. She seemed to understand and know about this limitation as it exists within much of current modern psychology. Hence, she refrained from using the word "psychology" in an isolated sense, always adding "depth" to the word or, in speaking about the human psyche, other terms which refer to that which is subliminal, subconscious, or unconscious. In addition, or as a corrective, she wanted to create a Buddhist version of depth psychology in terms now of its being a "science of mind" as a religious psychology. This is fundamentally crucial if we are to understand and to accept a claim which speaks about depth psychological elements within her life and dharma talks. In this sense, Seon Master Daehaeng could be regarded in contemporary Korean Buddhism as a pioneer, as the author of the first works which can be only understood properly if one's point of departure is the kind of meaning revealing depth psychological elements.

**Funding:** This research received no external funding.

**Institutional Review Board Statement:** Not applicable.

**Informed Consent Statement:** Not applicable.

**Conflicts of Interest:** The author declares no conflict of interest.

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
