# Peer review of "Depth Psychological Elements in Seon Master Daehaeng’s Dharma Talks, with Special Reference to Hanmaum Yeojeon"

_religions, doi:10.3390/rel12020114_

Round 1
Reviewer 1 Report
The author has chosen an interesting topic that needs more polishing and focused reorganization to make its points (if this reviewer adequately comprehends the points) more effectively.
1) The Introduction takes too long to get to the essay's subject, which should be upfront, then proceed to a rearranged format that perhaps better contextualizes the two research streams -- Buddhist Daehaeng vs. Depth Psychologist Daehaeng -- in terms of both existing research and broader questions. As the author points out, the approaches need not be contradictory, but to accept their complementarity implicitly assumes the more universalist approach apparently rejected by the Daehaeng-as-Buddhist school. Is this simply another example of a broad religious thinker forced into a narrower, more manageable and teachable scope by followers more interested in promulgating and filling out a tradition than in broad spiritual truths, along the lines of the familiar argument that Christianity was created not by Christ but rather by Paul, or, on a slightly more secular scale, Karl Marx's admission that "je ne suis pas marxiste"? Or is Buddhism, and perhaps East and South Asian religion more broadly, particularly prone to this tension? Is Daehaeng, as the author might be briefly suggesting at one point, the formulator of a content-less Korean Zen Buddhism, the form that has shaped Western attitudes over the past three quarters of a century?
2) The argument, it seems to this reviewer, needs to be tightened, since there are really two separate issues involved that are not always clearly distinguished. One is that Daehaeng's personal experience exemplifies and perhaps helps explicate the psychological approach pioneered especially by William James as well as later depth psychologists; the other would attest that her experience in fact constitutes her theology. It is the latter that would be the more radical argument, and I believe the one that the author wishes to argue. Is is also, of course, the form that would be far more antithetical to the Daehaeng-as-Buddhist approach, which is why the argument must be clearly stated and situated vis-à-vis the Introduction.
3) Closely involved with the need to better highlight crucial aspects of the argument is the need to excise less relevant material that fuzzes the argument's clarity and unbalances the article's organization. The potted history of modern psychology is largely not necessary, particularly if the particular depth psychologists of interest to the author were introduced earlier, in the Introduction.
4) The English would still need work before the piece is publishable. In some cases this reviewer wasn't sure of the author's meaning. What, for example, is a mantle in front of a tomb (p. 4, lines 124-125)? I would suggest that after revisions the author show the manuscript to a native-English-speaking scholar for polishing before resending it.
The issues raised in the piece extend beyond the life, thought, and legacy of one spiritually nourishing woman in postwar Korea. I hope the author can quickly make the necessary revisions to allow the journal to share his insights in print with the global research community.
Author Response
Dear Reviewer,
Thanks for your suggestions for the further refinement of my paper. I do really appreciate them. They are very helpful for the rethought of my paper. I revised them according to your suggestions as follows:
1. I reduced the introduction part and clearly mentioned why I did write this paper. I deleted the main part of the introduction argument of two trends of research on Daehang's works. You could see it in the revision paper: lines 17-45.
2. As you suggested, I did also tighten my argument that Daehaeng would be regarded as a originator or initiator of Korean version of depth psychology. I did it in abstract and also in the last section of this paper.
3. Also I deleted most of historical parts of psychology in the paper in the sections of 4, 5.
4. Your question about a "mantle" : I rewrite it as "tombstone".
Reviewer 2 Report
The article requires extensive revisions. First, it concerns almost exclusively psychological issues, and their religious interpretation is rather unconvincing. On the top of that, Author operates with an extremely shallow understanding of psychology ("Unfortunately however today, modern psychology, as a consequence of its empirical methodological orientation, does not actively promote this dimension even if we should want to move toward a deeper understanding of the human mind or psyche" or "If we go back and look at Daehaeng's usage of terms, we can easily think that, in her language, similar meanings are being meant as we find this in how Freud speak about the unconscious"). Second, the article is too biographical in its character, which translates into its uncritical attitude towards the thought in question. It is very difficult to recognize its main thesis; furthermore, the historical and social context of the thought is also missing, which includes the key concept of Appa, introduced without any further explanation, as if the text started at page 20. Thirdly and most importantly, the language of the article either makes the statements almost impossible to understand or, in other cases, makes them basically trivial ("In her life, on several occasions that were both pivotal and foundational, her potential feeling and imagination underwent changes in shifts which moved into instances of actual knowledge although, at every step along the way, her experiences and receptions of understanding and knowledge were never exhaustive" or "Simply put, we need to turn things over to the life and being of Juingong").
Author Response
Dear Reviewer,
Thanks for your comments on my paper. I am now telling you my thoughts on your comments in relation to my paper.
1. You commented that the author's understanding of modern psychology would be extremely "shallow". Unfortunately I do not understand your comment on it in relation to Freud. My understanding of the history of psychology seems to be different from yours.
2. You commented that my article is too biographical. Yes that is one of my focuses on Daehaeng's dharma talks. I attempted to let her talks to be telling as a sort of Korean Buddhist version of depth psychology. This is "original" part that other commentators mentioned. In this part I am wondering whether you would see my main focus.
3. You argued that the writer's English is problematic. I do accept your comment. Yes I am now native English speaker. Nevertheless I expected more concrete comments on the paper for its further development.
Anyway I do appreciate your comments!
Reviewer 3 Report
Dear Authors,
Thanks for your article. This is a relevant contribution whose content is not well known. It is original and it joins elements of Buddism and the Psychology of Religion. However, some elements should be improved.
From a general point of view, please, review the extension of some sections. Frequently they are very long and it may be better to create subsections (see particularly sections 4, 5, and 6, but also the section 2.
The number of Keywords have to be reduced. The “Instructions for Authors” of this journal indicates the following:
“Keywords: Three to ten pertinent keywords need to be added after the abstract. We recommend that the keywords are specific to the article, yet reasonably common within the subject discipline”.
You should read all the keywords and select only the most representatives.
Line 20. An edited work of 800 pages titled Hanmaum Yojeon is mentioned but any reference is cited in the text. The reader should find a reference in the list of References of the article of this work and, on the other hand, that reference should be included in this paragraph.
Before of this, because of the international approach of this journal, as well as the fact that the mind character of this article (ie Seon Master Daehaeng) is not universally known, a short biography and notes about her relevance in her own area of knowledge, should be given at the beginning of the article. In this case, in the Buddism. Some information could be given about the previous publication of works about Daehaeng in Buddism, Psychology of Buddism, Hystory of Buddism, etc. This exposition, in the first section, could conclude with a description of the present situation, the background and approach of this article, its objectives, and a general justification of them.
Notepage nº2. This note should include the reference of the two publications mentioned (only author and year), and the full reference be included in the list of references at the end of the article.
The allusion to “most scholars favor this approach”, and the following sentence (“Buddisht scholars…” should include same examples. It could be mentioned for example:
“Most scholars favor this approach (see for example author, year; author, year). Several authors should be cited as support of each one of these affirmations.
Line 28: “Though most papers are titled with her name and her works…”. The second part of the sentence should be modified. It is not evident the meaning that the authors want to give to the expression used: “titled with her name and her works”. The reference to “her works” is confusing because of the meaning of the following sentence. The “name of her works”?
Again, examples of these works should be given to support the sentence: “most papers…” . It is necessary to add references between parentheses. Or at least, citing the reference where this affirmation was read from.
“Though most papers are titled with her name and her works (e.g. author, year).
Line 33: What the means of “a form of validity” is? What is validated by the Daehaeng`s work? The sentence should be reviewed.
Line 44. A reference is given where two pages are mentioned (ie. 2(Kim 2019, 257-258)” but none quote appear in the text.
Lines 44-45: Examples of works with focus on humanities and social sciences and other fields should be given at the end of the sentence to support it.
Lines 54-58: “However, no one can easily deny the fact that, by attending to the explicit aspects of her work, one would be aligning oneself with the aforementioned second type of research approach which is given the extent of Daehaeng’s concern for the current situation of our human lives and asking questions about the future emergence of our human life in a transition that moves from our present situation toward what could be our new future situation”.
This last sentence is very subjective and requires some type of explanation. Why “no one can easily deny” the exposed idea in this paragraph?
In line 69 is told that, as a result of her “religious conversion” had a result “a new religious foundation”. The content or characteristics of these foundations should be described in the text. This should be completed with a clarification of the reasons to explain why her conversion was not conventional.
In addition, and to end the Introduction section, It should be mentioned what will be the contribution of this study (of the three aspects mentioned) to the development of knowledge.
The title of the second section is not one directly related with the contents previously presented in the Introduction section and its relationship with one of the three points is not evident. This is because the end of the introduction should finish with a preview of the contents (or sections or subsections) that will be developed in the following sections.
This second section includes many references to biographical information. That information should be supported with references. For example lines 99-100, 103, 105-107, 113-115, 119. It is not clear if the source is Daehaeng (1993). If this is the case this reference should be previously cited and, indirectly, do reference to it where necessary, or repeat if the mention is in different paragraph.
Again, in line 136 we find a page included in the reference but there is not a quote. The page should be only indicated in case of a quote. This happens again in lines 142 and 149.
The idea about the “massive capacity for empathy” of Daehaeng should be supported with some information and a reference, because is told that she, “from an early age, manifested such a capacity for empathy”. If this is the case, it should be not difficult to cite the source and expressions of her empathy.
With regarding to the inclusion of quotes, these should be included in the text more frequently in order to sustain the frequent allusions to psychological and personal characteristics of Daehaeng that appear in this second section. Verbatim fragments of her works should be included in the writing for the purpose of sustaining and giving credibility to the different affirmations about the life and personality of Daehaeng.
The third section. It would be interesting to give information about theoretical approaches alternatives to the psychological perspective, although only this last be further developed (lines 178-179).
Section 3 describes how Daehaeng built her playground. From the beginning to the line 242 this is described in detail. The reasons for this, following the reasoning showed in the text bases on the Winnincott’s notion of playground is interesting and well developed. In line 228 a new idea is introduced referred to the extension of the notion of playground to the remaning years of her life. What moves Daegaeng to expand her playground beyond the childhood? Is only the presence of nature? This should be inquired in the text. It is not clear in the text if the Buddisth scriptures played a role in that process.
I think that the text “he received” in line 255 have to be substituted by “she received”. Please, write the correct word.
The sentence in line 262 should be supported with a reference as was done with the first, second, and fourth examples of expansion of Daegaeng’s notion and sense of playground: Third, she expanded her playground to include a cosmic or an invisible world. No unknown thing is to be excluded but, instead, be actively embraced.
From my point of view, the authors should examine whether the concept of “religious experience” is well used in lines 306 to 313. From my point of view the discourse used by the authors to describe the experience of Appa as something gradual vs sudden depending of its relationship with the experiences lived before of that moment is not correct, or at least, is confusing. This is because the concept of gradual versus sudden process that is described is more related, from my point of view in this case, with the concept of conversion, that would be gradual or sudden, instead of the (religious or spiritual experience). I invite the authors discern what concept is better in this case and re-writing the paragraph in a way appropriate for this situation (ie. lines 306 to 313).
Could be inserted in the text any reference to support the following affirmations? (these are examples but there are many references in the article that seems to be subjectives and unsupported, most of them referred to biographical aspects (these should be reviewed throughout the text):
Lines: 368-369: “She could not stay comfortably at home because of her mother’s obstinate requests that she should get married”
Line 370-371: “she indicated that she was quite uninterested in getting married”
Line 394: “She cut her hair and she continued to yearn and search for a knowledge of Appa”
Lines 402-405: “Although she was initiated to be a monk, officially, she did not belong to any temple. She continued to live in a forested area for the sake of her religious quest in order to come to a fuller knowledge of Appa. However, she often had to return the daytime to care for her family and run a small shop”.
Lines 426-428: “She thought and speculated in relation to her experience that if father were to come to her son’s tomb, he would become her son and, conversely, if her son were to come to her father’s tomb, he would become her father”.
Many readers unknown the concepts of Appa and Juingong. The comprehension of the text, and its openness to a higher number of readers, would be improved including the description or definition of these concepts.
Redarding the two paragraph composed by lines 467 to 490 when elements of the history of psychology are presented in order to introduce the use of concepts of depth psychology, these should be reduced to the presentation, and definition, of the concepts needed for developing the use of these concepts by Daehaeng.
The additional information about academic and historical issues should be removed. A second option should be removing these paragraph and indicating the similarities between the definitions of reference (with references supporting the discourse) and the use of these concepts by Daehaeng. The discourse should be on similarities and dissimilarities between Daehaeng’s use of these concepts, and the definitions (and authors) of reference. Probably the exposition of the ideas could be improved with the presentation of the Daehaeng conceptualization of the reference concepts and then, the similarities and dissimilarities with relevant theoretical approaches and authors. It makes no sense to develop in detail the three school of psychology in the text if the ideas and concepts used by different authors are not related with the works, ideas, conceptualization , and developments of Daehaeng. This fifth section should be reviewed in detail removing the not relevant information. These could be also relocated.
The sixth section, whose title is “The Function and the Image of Juingong” is well structured and well written but without support references, but due to the different lines of content that are developed it would better dividing it in subsections. It has to be reviewed and the sentences that requires be supported have to be rewritten in order to avoid presenting information that might seem subjective. This is especially relevant from line 556 to 583, 638 to 639, 644 to 659, 673-683.
Likewise, the references to terms of depth psychology should be associated with specific theoretical approaches and concepts (see lines 628 to 643).
The third image (ie telephone or mailbox) need to be associated with a reference as was done with the first, the second, and the fourth image.
Direct or indirect support should be given to the following sentence (line 737 to 743): This energy exists in the nature of all beings as a life giving thing, the nature of a given thing pointing to the kind of energy which properly belongs to it. Hence, if we are to renew the life of our psyches, we must attend or allow for a form of energy that is joined to foundations in the unconscious. It is an unconscious energy which Jung identified in terms which speak about a numinosum given how Otto speaks about the sense of bonding and the circulating energy between an ego and an unconscious which is joined to one's ego. References of Jung and Otto should be inserted in the text.
The same happens with lines 744 to 750: it requires support references, especially with the mention to “depth psychologists” who “speaks about….”. After with “Between the place and being of Juingong and our subconscious, an intimate relation exists. One is found in the other or one points to the other”.
The seventh section, conclusions, is too extensive to be a last section with the mind points of the article and, possibly, some suggestions with a prospective character given the approach of the article. This section should be reduced and all the information not suitable for this section should be removed or moved to others section (or subsections) more appropriate.
Author Response
Dear Reviewer,
Many thanks for your concrete comments and suggestions. I do really appreciate them. They are very helpful for the further development and refinement.
According to your suggestions, I revised my paper as follows:
1. You suggested to make subsections in each section of my paper, especially sections, 2, 4, 5, and 6. I revised them in each section with subsections.
2. The number of Keywords would be reduced. I reduced half of them.
3. Short biography would be introduced in the introduction part. And I deleted the two research trends in the introduction part. And I rewrite my introduction according to your suggestion.
4. I corrected "he received" as "she received" in line 258.
5. I did add references that you asked for the support of my paper as follows: Lines 99-100, 103, 105-107, 113-115, 119, 142-149, 178-179, 306-313, 368-369, 370-71, 394, 628, 737-743, 750.
6. In conclusion, I reduced much information and pointed out key prospective for the article.
Once again I do appreciate your considerate comments of my paper.
Round 2
Reviewer 2 Report
The main argument has been clarified and the most dubious section have been removed. Although I am still not convinced whether such a case study would be of general interest to other religious scholars, the paper itself does not raise any major theoretical objections, my evaluation of its latest version is positive.
Author Response
Dear Reviewer,
Thanks for your considerate reading of my revised essay. I am encouraged to hear that you see the author's point. As you mentioned, I recognize that this essay does not aim at creating a brand new theory for the depth understanding of Korean Seon Master's works. Rather it would attempt to suggest a possibility for a depth psychological interpretation of her works.
Once again I do appreciate your two reviews for the deeper revision of this essay.
Thanks,
Reviewer 3 Report
Dear Authors,
Thanks for your considerations and changes made to your article. Although the original text has been significantly modified, additional suggestions can be made because much of the problems that are commented in the following lines are now more evident that in the first version of the article.
After reading the text, I would like to suggest to the authors additional recommendations.
The abstract contains the following sentence: “This essay hopes to reveal why she should be regarded as the originator of a Korean Buddhist version of depth psychology”. The text of the article is not a deep analysis of the similarities and presence of elements of deed psychology in the document. This affirmation should be rewritten and tempered.
The same should be done with the sentence in lines 111 and 112: “This essay hopes to reveal why she should be regarded as the originator of a Korean Buddhist version of depth psychology”.
The end of the Introduction of the text should include the objective of the essay as well as an advance of following sections that will be developed in the text and their goals.
The fourth note should be removed (number four inserted in the text in line 125).
The second section should start with a justification about the relevance of the Daehaeng’s childhood in her works or legacy.
Lines 129 to 139 should be reviewed. The writing is confusing and it is observed a repetition of words in the same paragraph with the same meaning.
The reference that support the following sentences should be added to the text:
Line 141: “Although she was born into a stable family environment, at age 7 she had to join her family, in fleeing their urban residence to escape from the efforts of the Japanese Imperial police to arrest her father”. The same happens with lines 147 to 152. If the source for this information is Daehaeng, 1993, 22, this reference should be moved to the beginning of the paragraph before of giving the information showed from line 141 to 152.
Could be possible to insert in the text some reference or literal reference to support the discourse given in lines 194-201?
Section 3. In my opinion, the title does not show the content of the section. Probably, a title as “The Playground in the Years of Childhood and Beyond: a psychological interpretation“, or similar, would be more appropriate. The present title fails to reflect the content of the section because this is not centered on psychological implications but in giving a psychological interpretation of different ideas based on the Daehaeng’s works.
The lines 220 to 222 should mention and specify that the content of this section is centered on the psychological implications of the playground.
It would be necessary to mention the work of Winnicott 1971, 1-3 at the begging of the last paragraph starting in line 249. For example:
Line 249: “According to Winnicott (1971, 1-3), there things are fundamental…”
Line 275: The following sentence need to be supported with a reference: “To expand the size of her playground, she began to live in other mountainous areas”.
Section 4. The “more” word, as is used in line 315, and in this section, probably is better referenced as a substantive. This means that the writing of the line 315-316 would be better in the following way: “constituent elements, the “more” dimension”, and religious life transition”. The same modification should be done in the title of the second subsection: “2) The “More” dimension”.
The following sentence needs to be supported with a reference: “religious life does not exist as some sort of magic, apart from our need to be personally active and involved”. If the reference of this sentence is “(Corbett 2015)”, this should be previously inserted in the text, at the beginning of the paragraph (line 322).
The following sentence needs to be supported with a reference: “Prior to her first experience of Appa, the locus of Appa existed only as a potentiality but, after her first experience of it, it became a new center in a displacement which marginalized her past place and condition. This change, as the most important moment of awakening for her in her life, encouraged her to try and move toward a more intimate knowledge of this “more dimension” as this refers to the being and reality of Appa”.
The same happens with: (line 431): she could not find consolation that she desired that, in some way, resembled the power of her initial experiences”. This sentence needs to be supported with a reference.
The title of the fifth section is confusing and more extensive that the content that the reader find. This should be more realistic, as for example something like this:
“The presence of elements of depth psychology in the Image of Juingong in Daehaeng works”.
It would be interesting to qualify the scope of this section correcting some sentences because is not appropriate to consider the section as a rigorous analyses of the Daehaeng’s works from the depth psychology perspective. I suggest the following corrections:
Line 545-546: “…can be aligned to some theoretical and practical approaches specific to the field of depth psychology”, instead of “can be aligned to differentiations that we will find within the theory and the practice of depth psychology”.
Line 530: “that belong to the field of the so-called deep psychology.” Instead of “that belong to depth psychology”.
Lines 719-720: the meaning of the sentence is not limited to the depth psychology area. I suggest to write “More specifically, Daehaeng’s three attitudes of life are connected to the therapeutic function of psychology as a means that can help us move toward a greater growth” instead of “More specifically, Daehaeng’s three attitudes of life are connected to the therapeutic function of depth psychology as a means that can help us move toward a greater growth”.
The first subsection (ie.1) Life Attitude) does not includes any reference. Although the discourse looks like a description or and interpretations, this should be supported with quotes, or, at least, mentions in the text to specific works of Daehaeng (for example “see Daehaeng, 1993, 1-4). This should be significantly improve throughout the subsection. An essay without that references would seem to lack objectivity and, on the contrary, it would seem (above all) subjective and unjustified. This should be avoided and the text modified. Please, review it. This is especially evident from the beginning of the section to the line 669.
In addition, previously was mentioned that the way in which some elements of depth psychology are presented is troublesome, and now can add that sometimes is also questionable. This is especially relevant in some parts of the text:
For example:
Lines 720 to 724: “According to the insights of depth psychology and the teachings of classical psychoanalysis our mind or psyche manifests symptoms of illness just as the body does, although in terms of mental illness we need to transcend any conditions or states of repression within and below our conscious existence”. The only way of maintaining this sentence in the article would be in form of a quote, with a text taken from an original source that have to be cited in the text mentioning author, date of publication and pages where the text can be found.
The same happens with lines 725 to 730: “Through a form of psychotherapeutic conversation that exists between a therapist and a client, a counsellor and a counselee, or an analyst and an analys and, these repressions can be identified and so, in some way, released or expressed although, in this process, resistance from a client, a counselee, or an analys and will impede further development and healing if, in some way, further conversations or dialogue is prevented or impeded. Conversely, an oversight on the part of a therapist, a counselor, or an analyst, will lead to problems that will arise from misdiagnosis”.
And is more evident in the third section (ie. “3) Depth Psychological Image), where is assumed the universal knowledge of the four images mentioned by the readers. Each image should be related with its corresponding reference (for example a specific author and work where that image is described). If this is not done, it is not possible to sustain the correspondence between the four images that the author find in the Daehaen’s works and the cited psychological images.
Other additional example is in lines 845: “where Daehaeng speaks about a religious energy in our minds or the place of Juingong, depth psychologists speak about an energy that somehow comes from our subliminal subconscious, unconscious realm. Between the place and being of Juingong and our subconscious, an intimate relation exists. One is found in the other or one points to the other. In this sense, we can claim that Daehaeng should be regarded as initiator of a Korean Buddhist version of depth psychology”. The sentence affirming that “an energy that somehow comes from our subliminal subconscious, unconscious realm” needs to be supportd with references.
The most important and questionable idea is the conclusion that “Daehaeng should be regarded as initiator of a Korean Buddhist version of depth psychology”. The conclusion is not supported by the development of the text, and we can find different reasons that give support for this problem. For example, the article did not give information about the development and history of the Depth psychology in Korea, and, as was previously told, the information showed in the text about this psychological area is not well supported and it is not representative. In addition, other elements from the psychological analysis of Buddhism should also be given previously.
The best solution is, probably, to delete these paragraphs and center the discourse in the description of the presence of the therapeutic function in the Daehaeng’s works. Again, some references would have to be included from lines 741 to 750, and 764 to 775. The Discourse is very subjective and references have to be added to support the different affirmations given in those lines. It is possible that the only conclusion that could be given is that, according to the author of this article, Daehaeng could be the first Buddhist author in which works is possible to find elements of depth psychology.
The title of the article would have to be modified with a more qualified approach.
The last section (ie. Conclussions) have to be reviewed and corrected following the suggestions made related with the way in which the depth psychology is considered.
As the author can see, a deep rewritten of the text is required.
Author Response
Dear Reviewer,
First of all, I would like to say "many thanks" for your exact comments on my essay. They have been very helpful for it to develop further. I followed your comments precisely for the revision of my essay.
1. I revised the introduction part on the objective and the development process of this essay.
2. I added my justification of the importance of Daehaeng's early life for her religious life.
3. I toned down my argument of Daehaeng as you suggested. So I replaced the words, "the originator of a Korean Buddhist version of depth psychology" as the first author of works, a pioneer"...
4. I changed the title of the fifth section.
5. I revised other things that you requested to revise..
Once again I do appreciate your straightforward comments!
Round 3
Reviewer 3 Report
Dear Authors,
Thanks for your attention and efforts to improve the article. I think that it meets the minimum requirements to be published but It would be better with a best citation and contextualization of the biographical information mentioned in the body of the article.
Author Response
Dear Reviewer,
Thanks for your another request to revise. Yes I add the contextualization of my essay as follows: "This essay does not refer to Buddhist interpretations of Daehaeng's dharma talks. Instead, it tries to situate her dharma talks in relation to sources which point to elements of depth psychology." The reviewer seems to be Buddhism scholar. This essay is not a Buddhist interpretation! I hope that the reviewer would see my point in this essay. Thanks, Author.
Thanks.
Author